# A Custom Panel for Profiling Microglia Gene Expression

**DOI:** 10.3390/cells13070630

**Published:** 2024-04-04

**Authors:** Phani Sankar Potru, Natascha Vidovic, Susanne Wiemann, Tamara Russ, Marcel Trautmann, Björn Spittau

**Affiliations:** 1Bielefeld University, Medical School OWL, Anatomy and Cell Biology, 33615 Bielefeld, Germany; phani.potru@uni-bielefeld.de (P.S.P.); natascha.vidovic@uni-bielefeld.de (N.V.); susanne.wiemann@uni-bielefeld.de (S.W.); tamara.russ@uni-bielefeld.de (T.R.); 2Gerhard-Domagk-Institute of Pathology, Münster University Hospital, 48149 Münster, Germany

**Keywords:** microglia, BV2 cells, bone marrow monocytes, RNA, NanoString, nCounter technology

## Abstract

Despite being immune cells of the central nervous system (CNS), microglia contribute to CNS development, maturation, and homeostasis, and microglia dysfunction has been implicated in several neurological disorders. Recent advancements in single-cell studies have uncovered unique microglia-specific gene expression. However, there is a need for a simple yet elegant multiplexed approach to quantifying microglia gene expression. To address this, we have designed a NanoString nCounter technology-based murine microglia-specific custom codeset comprising 178 genes. We analyzed RNA extracted from ex vivo adult mouse microglia, primary mouse microglia, the BV2 microglia cell line, and mouse bone marrow monocytes using our custom panel. Our findings reveal a pattern where homeostatic genes exhibit heightened expression in adult microglia, followed by primary cells, and are absent in BV2 cells, while reactive markers are elevated in primary microglia and BV2 cells. Analysis of publicly available data sets for the genes present in the panel revealed that the panel could reliably reflect the changes in microglia gene expression in response to various factors. These findings highlight that the microglia panel used offers a swift and cost-effective means to assess microglial cells and can be used to study them in varying contexts, ranging from normal homeostasis to disease models.

## 1. Introduction

Microglia are the sentinel tissue-resident macrophages that are present in the parenchyma of the central nervous system (CNS) and play a vital role in the development, maturation, and maintenance of the CNS. Apart from participating in immune responses within the CNS, during which they attain a reactive state, they actively survey their microenvironment, modulate synaptic connectivity, and support neuronal development [1,2,3] Various factors carefully regulate all the mentioned characteristics of microglia and their responses to prevent microglia dysfunction [4]. Interestingly, microglia dysregulation was shown to contribute to the pathology of neurological conditions such as Alzheimer’s disease, Parkinson’s disease, multiple sclerosis, and neurodevelopmental disorders [5,6].

Recent advancements in RNA and single-cell sequencing analyses have provided valuable information regarding the gene expression patterns of microglia. Genes such as Transmembrane protein 119 (*Tmem119*), Olfactomedin like 3 (*Olfml3*), Purinergic Receptor P2Y12 (*P2ry12*), Spalt Like Transcription Factor 1 (*Sall1*), Hexosaminidase Beta (*Hexb*), G protein-coupled receptor 34 (*Gpr34*), Fc receptor-like S (Fcrls), and Sialic acid-binding Ig-like lectin H (*SiglecH*), etc. were shown to be uniquely expressed by microglia when compared to other cell populations [7,8,9]. Understanding this microglia-specific gene expression is crucial to disentangle their diverse roles during homeostasis and pathological states [10]. While the conclusions from carefully constructed studies have provided significant insights, there are hindrances to fully understanding the intricacies of microglia biology. Factors such as the heterogeneity of microglia, both spatial and temporal, for example, pose a considerable challenge in this endeavor and make a case for further microglia-specific studies.

To address this, we have designed a NanoString nCounter-based custom panel designed specifically for profiling murine microglia gene expression. Our custom panel incorporates 173 microglia genes implicated in their homeostasis, activation, phagocytosis, cytokine/chemokine signaling, lipid metabolism, and important mechanisms such as Transforming Growth Factor β1 (TGFβ1) signaling, and five housekeeping genes (Figure 1 and Appendix A). NanoString nCounter technology utilizes unique molecular barcodes that are assigned to each target gene, ensuring accurate and precise quantification of gene expression levels. This technology employs hybridization and direct digital detection, which allows for highly sensitive and reproducible measurement of RNA transcripts without the need for amplification steps.

To evaluate our panel, we used total RNA from bone marrow monocytes (BMMn), the BV2 microglia cell line (BV2), primary mouse microglia (pMG), and MACS-isolated microglia from 30-day-old adult mice (Adult MG). Our findings reveal a distinctive pattern of microglia gene expression. We report that homeostatic genes exhibit increased expression in adult microglia, followed by primary cells, but are notably absent in BV2 cells. Conversely, reactive markers displayed elevated levels in BV2 cells. Intriguingly, some of the homeostatic markers were found to be higher in bone marrow monocytes than in BV2 cells. Taken together, this study underscores the efficiency of our NanoString-based microglia panel in providing a swift and cost-effective multiplexed approach to assessing microglia cells based on their gene expression and can be employed across various disease models as well as in future comparative studies.

## 2. Materials and Methods

### 2.1. Animals

Postnatal day 0/ to 3 (P0 to P3) pups from pregnant NMRI mothers (Janvier, Le Genest-Saint-Isle, France) were used for establishing primary microglia cultures, while 30-day-old C57BL/6J mice (Janvier, Le Genest-Saint-Isle, France) were used for the isolation of adult microglia. Bone marrow monocytes were isolated from the femurs of adult NMRI mice (Janvier, Le Genest-Saint-Isle, France). The mice were housed at a temperature of 22 ± 2 °C, following a 12-h light/dark cycle, with unrestricted access to chow and water. All animal experiments were performed in adherence with the German Federal Animal Welfare Law and local ethical guidelines.

### 2.2. Primary Microglia Culture

Primary microglia culture preparation was performed as previously described [11]. Briefly, brains from NMRI pups at P0/P1 were rinsed with Hank’s balanced salt solution (240201117, Thermo Fisher Scientific, Dreieich, Germany), followed by the removal of meninges and blood vessels. The dissected brains were then immersed in ice-cold HBSS and subjected to digestion using 1X Trypsin-EDTA (25300054, Invitrogen, Darmstadt, Germany) for 10 min at 37 °C. Then, ice-cold fetal calf serum (FCS) and DNase (M0303S, New England Biolabs, Frankfurt am Main, Germany) were added in equal amounts at a final concentration of 0.5 mg/mL before brain dissociation with serological pipettes of increasing size. The dissociated cells were centrifuged, and after discarding the supernatant, the cell pellet was re-suspended in DMEM/F12 medium (11320033, Gibco, Schwerte, Germany) supplemented with 10% FCS (F9665, Sigma-Aldrich, Schnelldorf, Germany) and 1% penicillin/streptomycin (P06-07050, PAN Biotech, Aidenbach, Germany). Finally, the cell suspension was seeded into poly-L-lysine-coated (P2636-25MG, Sigma-Aldrich, Schnelldorf, Germany) tissue culture flasks at a density of 2–3 brains per 75 cm^2^ flask. Mixed glia cultures were washed for two consecutive days following the initial plating and were then allowed to grow for two weeks. Pure microglia were separated by tapping the flasks. Cells were then collected and plated into 6-well plates for further analysis.

### 2.3. BV2 Culture

The BV2 mouse microglia cell line was maintained in DMEM/F12 (11320033, Gibco, Schwerte, Germany), supplemented with 10% heat-inactivated FCS (F9665, Sigma-Aldrich, Schnelldorf, Germany) and 1% penicillin/streptomycin (P06-07050, PAN Biotech, Aidenbach, Germany). Cells were cultured at 37 °C in a 5% CO_2_ and 95% humidified atmosphere in 75 cm^2^ flasks. For plating, cells were washed with 1X PBS and dissociated from the flask by trypsin (SLCN8219, Sigma-Aldrich, Schnelldorf, Germany). Then, they were pelleted and resuspended in a culture medium and plated into 6-well plates for further analysis.

### 2.4. Bone Marrow Monocyte Extraction

Bone marrow-derived monocytes (BMMns) were collected using the protocol reported by Wagner and colleagues [12]. Briefly, the femurs of NMRI mice were dissected, disinfected with 96% ethanol, and perfused with 4 mL of PBS. The suspension was collected in a 15 mL tube and centrifuged. Then, the cell pellet was resuspended in 2 mL of red blood cell lysis buffer 1X (00-4333, Invitrogen, Darmstadt, Germany), moved into Eppendorf tubes, and incubated at 4 °C for 10 min. Upon centrifugation, the cell pellet was directly subjected to RNA extraction.

### 2.5. Ex Vivo Microglia Isolation and Flow Cytometry

Microglia from adult C57BL/6J mouse brains were isolated using the Adult Brain Dissociation Kit (130-107-677, Miltenyi Biotec, Bergisch Gladbach, Germany) and MACS technology according to the manufacturer’s protocol. Briefly, mice ≥ p7 were sacrificed, and the brain tissue was carefully dissected. The brain tissue was dissociated using a gentle MACS Octo Dissociator with heaters, and cells were filtered through a 70 μm cell strainer to obtain a single-cell suspension. Debris removal and red blood cell lysis were performed as described in the protocol. Subsequently, microglia were magnetically labelled using murine CD11b MicroBeads (130-126-725, Miltenyi Biotec, Bergisch Gladbach, Germany) and separated using MS Columns (130-042-201, Miltenyi Biotec, Bergisch Gladbach, Germany) on an OctoMACS separator. Cells were filtered through a 30 µm pre-separation filter before being applied to the columns. The positively selected CD11b^+^ microglia were eluted, yielding a highly enriched microglial population. Cells were analyzed using MACSQuant 10 and MACSQuantify Software V2.13.3. In short, cells were stained with anti-mouse CD11b-Vioblue (130-113-810, Miltenyi Biotec, Bergisch Gladbach, Germany) at 1:50 in flow cytometry buffer for 10 min. Afterwards, cells were washed and applied to the flow cytometer. First, cells were gated as single events and then according to CD11b^+^ fluorescence (Figure 2b).

### 2.6. RNA Extraction

Total RNA extraction from primary microglia, BV2, BMMn, and adult MG was performed using TRIzol (15596026, Invitrogen, Darmstadt, Germany) according to the manufacturer’s instructions. RNA concentrations were analyzed using the BioPhotometer D30 (Eppendorf AG, Hamburg, Germany). Then, equal amounts of RNA were diluted in ddH_2_O and frozen at −80 °C until NanoString analysis.

### 2.7. NanoString nCounter Analysis

RNA was analyzed on a NanoString nCounter^®^ system for microglia custom codeset (NanoString Technologies, Seattle, WA, USA) containing 173 genes of interest and six housekeeping genes as per the manufacturer’s instructions. Briefly, the reaction mixture containing the RNA sample, reporter probe, and capture probe was hybridized overnight at 65 °C. Then, the samples were moved into NanoString nCounter^®^ SPRINT cartridges for the removal of unbound probes and immobilization. Then the number of barcodes for each target was measured by a scanning microscope. Resultant RCC files from nCounter were then imported into Rosalind (OnRamp Bioinformatics, https://www.rosalind.bio (accessed on 27 November 2023)) for normalization. The normalized counts (Appendix A) were then uploaded into integrated Differential Expression and Pathway analysis software (iDEP) (http://bioinformatics.sdstate.edu/idep/ (accessed on 27 November 2023)) (v.90 and 1.1) to perform further analysis, including visualizations (Heatmap, PCA plot, and transformed expression plot) and identification of differentially expressed genes (DEGs). For DEG analysis, the criteria of FC >2 or <2 and FDR of 0.05 were used [13]. Volcano plots were prepared using SRplot [14]. Venn diagrams were made using the tool available at https://bioinformatics.psb.ugent.be/webtools/Venn/ (accessed on 10 January 2024).

### 2.8. Data Availability

Data analyzed in this study were downloaded from the public database Gene Expression Omnibus (GEO) from GSE79898 and GSE80304 [15] for lipopolysaccharide (LPS)-treated BV2 and pMG cells, while the data for TGFβ1-treated primary microglia cells was accessed from GSE115652 [16].

**Figure 2 cells-13-00630-f002:**
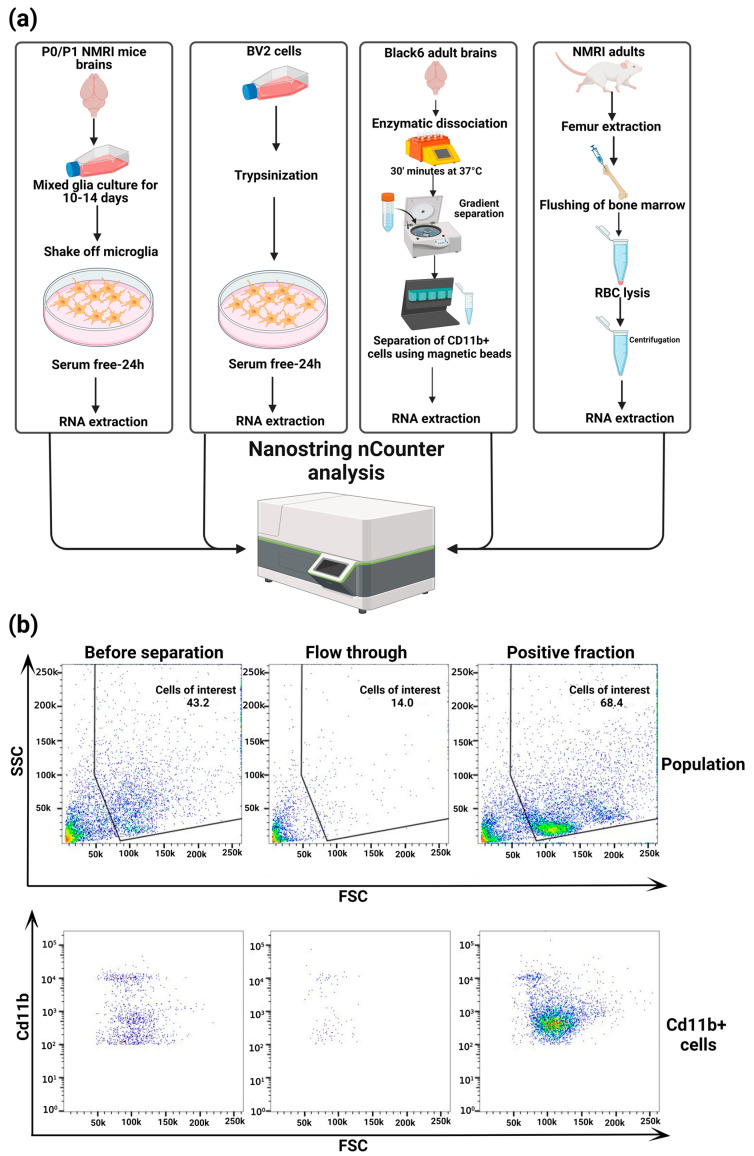
Experimental design and the gating strategy for adult MG separation using CD11b^+^ magnetic beads: (**a**) Graphical representation of experimental procedures involved in the RNA extraction and subsequent Nanostring analysis of adult microglia, primary microglia, BV2 cells, and bone marrow monocytes; Created with BioRender.com (accessed on 18 March 2024). (**b**) Gating strategy for flow cytometry and microglia enriched populations using CD11b magnetic beads. Flow cytometry plots of the total population of cells and CD11b^+^ cells before and after magnetic separation and flow through obtained during magnetic separation.

## 3. Results

### 3.1. Custom Panel Identifies Distinct Gene Expression Patterns in Microglia Cells

To identify microglia gene signatures using the custom panel, RNA from Adult MG, pMG, BV2 cells, and BMMn cells was subjected to NanoString nCounter analysis. Genes were selected based on the outcomes of recently published works that determined microglia gene expression under health and disease and in distinguishing them from monocytes/macrophages [10,17,18,19,20,21,22].

Principal Component Analysis (PCA) was performed to investigate the differences and similarities among the cell populations used. The PCA results revealed that PC1 accounted for 55% of the total variance, while PC2 explained 23% of the total variance (Figure 3a). K-means clustering was performed using the elbow method (Appendix A, Appendix A) on the gene expression data from the panel to identify distinct gene expression patterns among adult microglia, BMMn cells, primary microglia, and BV2 microglia. The analysis unveiled five clusters, each characterized by a unique set of genes with specific expression patterns. Cluster 1 is primarily represented by adult microglia and exhibited a significant enrichment of genes such as *P2ry12*, *Tgfbr1*, *Hexb*, *Olfml3*, and *Tmem119*. These markers were also expressed in primary microglia but were notably absent in BMMn and BV2 cells (Figure 3b,c).

Cluster 2 displayed distinct gene expression in which the genes are present in BMMn cells and adult MG while absent in primary and BV2 microglia cells. Notable genes present in this cluster are *Ccr2*, *Ccr3*, *Cd72*, and *Cd74*. Cluster 3 is characterized by genes such as *Lgals3*, *Fabp5*, *Nos2*, *Mif*, and *Lilrb4*. Genes that are significantly enriched in pMG and adult MG constitute cluster 4 and are represented by genes such as *Apoe*, *Jak1*, *Stat1*, *Stat2*, *Axl*, and *Mrc1*, suggesting their potential roles in distinguishing the functions of ex vivo microglia and primary microglia (Figure 3b,c).

Cluster 5 comprised genes such as *Spp1*, *C3ar1*, and *Gpnmb*, which were present across all microglial populations but absent in BMMn cells (Figure 3b,c). In general, homeostatic markers show a distinctly higher expression in adult MG, followed by primary microglia, while reactive state markers are present in in vitro microglia (pMG and BV2) (Figure 3d,e, Appendix A).

### 3.2. Differential Gene Expression across Microglia Cell Types and BMMns

Differentially expressed genes (DEG) in all the microglia cell types and BMMn cells involved in this study were assessed using iDEP (Figure 4a). Our findings revealed that, in adult MG compared to pMG, BV2 cells, and BMMn cells, 90, 130, and 113 genes were upregulated, respectively. Interestingly, a commonality among these upregulated genes in adult MG is that they are all known homeostatic markers such as *Csf1R*, *Cx3cr1*, *Gpr34*, *Hexb*, *Olfml3*, *P2ry12* and *13*, *Tmem119*, and *Tgfbr1*. Conversely, there were 25, 13, and 11 genes that exhibited downregulation. Among the downregulated genes in adult MG when compared to pMG were *Spp1*, *Thbs1*, *complement C3*, *Gpnmb*, and genes related to bone morphogenetic proteins such as *Bmp4* and *BMP7*. When compared to BV2 cells, adult MG displayed reduced expression of genes such as *Lgals3*, *Lilrb4*, *Lpl*, and *Mif*, while they showed lower expression of, among other genes, *complement C3*, and *Thbs1*, a change they share with pMG (Figure 4b, bottom panel). When comparing pMG to BV2 cells and BMMn cells, the DEG analysis demonstrated an upregulation of 95 and 73 genes, respectively. Primary microglia also expressed higher levels of most all of the homeostatic markers when compared to BV2 cells and bone marrow cells. These results highlight the inherent differences between the pMG and microglia cell lines, such as BV2 cells. There were 27 and 32 genes that were downregulated in pMG when compared to BV2 cells and BMMns, respectively. Genes such as *Cd74* and *Cd72* were commonly down-regulated between both comparisons, while genes such as *Lilrb4*, *Lgals3*, *Nlrp3*, and *Mif* were specific to the pMG vs. BV2 group. Similarly, when BV2 cells were compared to BMMn cells, 36 genes were upregulated, and 64 genes were downregulated. Interestingly, among the downregulated genes were the microglia markers such as *Cx3cr1*, *Gpr34*, *P2ry12*, and *P2ry13,* etc, while markers that were down in adult MG and pMG when compared to BMMns such as *Lilrb4*, *Lgals3*, *Lpl,* and *Spp1* were upregulated in BMMns when compared to BV2 cells, again emphasizing the caution required when drawing conclusions from the studies using BV2 cells (Figure 4b, upper panel).

Then, to check for specific and common changes in gene expression between different microglia cells and BMMns, Venn diagram analysis was performed. Upon examining the genes that were upregulated in adult microglia, 62 genes were found to be commonly upregulated across all groups. Additionally, nine, four, and 17 genes were uniquely upregulated when adult MG was compared to BMMn cells, pMG, and BV2 cells, respectively. Furthermore, eight genes were commonly upregulated in adult MG when compared to both BMMn cells and pMG. In terms of comparisons between adult MG and BMMn cells Vs adult MG and BV2 cells, 34 genes were commonly upregulated. Moreover, 17 genes were commonly upregulated in adult MG compared to both pMG and BV2 cells (Figure 5a and Appendix A, Appendix A).

Regarding downregulated genes, there were no genes commonly downregulated among all groups. However, we identified seven, seven, and 17 genes that were uniquely downregulated in adult MG when compared to BMMn cells, pMG, and BV2 cells, respectively. Only one gene was downregulated in adult MG compared to both BMMn and BV2 cells. Additionally, five genes were commonly downregulated in adult MG compared to both pMG and BV2 cells. The comparison of adult MG with BMMn cells and pMG revealed three genes that were commonly downregulated (Figure 5b and Appendix A, Appendix A).

Primary microglia (pMG) showed 48 genes to be commonly upregulated when compared with BV2 and BMMn cells. While pMG showed 41 uniquely upregulated genes compared to BV2 cells, only two genes were uniquely upregulated compared to BMMn cells. Notably, BV2 cells exhibited upregulation of 13 genes compared to BMMn cells. Additionally, pMG and BV2 cells shared 17 commonly upregulated genes when compared to BMMn cells (Figure 5c and Appendix A, Appendix A). Among the downregulated genes, there were 18 genes uniquely downregulated in pMG compared to BV2 cells, while 38 genes were uniquely downregulated in BV2 cells compared to BMMn cells. No uniquely downregulated genes were observed in the pMG vs. BMMn group. Interestingly, 23 genes were commonly downregulated in both pMG and BV2 cells compared to BMMn cells, and six genes were downregulated in pMG compared to both BV2 and BMMn cells (Figure 5d and Appendix A, Appendix A).

### 3.3. Custom Panels Can Account for Functional Changes in Microglia

In the next step, to validate the suitability of the genes in the current panel for functional studies, publicly available data from GEO (Gene Expression Omnibus) databases GSE79898, GSE803304, and GSE115652 were utilized. The custom panel genes from all the genes in these studies were selected using the vlook-up tool and normalized using iDEP. Our results demonstrated that LPS-treated BV2 and pMG cells exhibited decreased expression of homeostatic markers, while reactive markers were upregulated in both cell types. Specifically, LPS led to a reduction in *Hexb* and *P2ry12* in both BV2 and pMG cells, while *Gpr34*, *Olfml3*, and *Siglech* were down in pMG. Interestingly, the RNA-seq data from Das et al. [15] further support the findings in Figure 3a, in which, under control conditions, homeostatic markers were found to be highly expressed in pMG cells compared to BV2 cells (Figure 6a).

Additionally, we analyzed our previously published microarray data of pMG cells treated with TGFβ1 for 24 h. The results confirmed an increase in the expression of homeostatic markers in TGFβ1-treated microglia. Markers such as *Tgfbr1*, *P2ry12*, *Olfml3*, *Gpr34*, etc. were upregulated, while reactive markers such as *Stat1*, *Stat2*, *and Nlrp3* were reduced in TGFβ1 treated group (Figure 6b). Interestingly, LPS treatment led to increased expression of the above-mentioned markers, suggesting a reactive phenotype (Figure 6a). These findings indicate that the genes included in our custom panel can adequately account for the functional changes associated with microglia in response to various stimuli.

## 4. Discussion

In the current study, we report a murine microglia custom panel consisting of 173 microglia-relevant genes based on NanoString nCounter technology [23]. To characterize the microglia gene expression, we have analyzed RNA from BMMn cells, BV2 cells, pMG cells, and adult MG. The expression profile of the panel genes among different microglia cell types demonstrated that adult MG displays higher amounts of homeostatic microglia markers when compared to other cell types used. This points towards the homeostatic nature of ex vivo microglia when compared to in vitro microglia. These results are in accordance with a previous report by Butovsky et al., in which they showed that genes such as *P2ry12*, *Cx3cr1*, *Olfml3*, and *Tmem119* were significantly enriched in adult MG when compared to microglia cell lines, monocytes, and other CNS cells such as astrocytes, oligodendrocytes, and neurons. In total, they report a set of 150 genes that are enriched in microglia, which suggests that microglia possess their own unique gene signature [7].

Interestingly, in the current study, genes such as *Lpl*, *Spp1*, and *Ccl4* were found to be higher in microglia cell line BV2 cells, followed by primary microglia, when compared to adult MG. This can be due to the fact that microglia in the culture conditions are in a reactive state when compared to in vivo microglia. Cadiz et al., have shown that microglia, when taken from their native environment, suffer a “culture shock” and their transcriptome differs from that of freshly isolated microglia. They defined a set of “culture shock” genes characterized by the expression of *Apoe*, *Lyz2*, and *Spp1*. Interestingly, *Spp1* was found to be highly expressed in BV2 and pMG in the current study, while *Apoe* was enriched in both BV2 cells and adult MG [24]. Moreover, genes such as *Lilrb4*, *Lgals3*, *Itgax*, and *Mif* are also up in the in vitro MG. Among these, it has been recently shown that BMMn also expresses *Lilrb4* and that it is not a microglia-specific marker. However, reactive microglia were found to be expressing more *Lilrb4* than homeostatic microglia [25].

A meta-analysis of microglia gene expression from various neurodegenerative models and aging has identified a highly consistent transcriptional profile of up-regulated genes in primed microglia in which *Lgals3* and *Itagx* were prominent. These changes confirm the reactive nature of in vitro microglia and that of in vivo microglia under diseased conditions [19]. Among other genes that are upregulated in primary MG are *Stat1*, *Stat2*, *Axl*, and *Mrc1*. We have recently shown that *Mrc1* expression is increased upon inhibiting TGFβ1 signalling in primary microglia [26]. Since in vitro microglia in the current study were cultured under serum-free conditions for 24 h before RNA extraction to nullify the effect of growth factors, including TGFβ1, it is not surprising to observe this change. Additionally, a direct comparison of BV2 microglia and BMMn showed that homeostatic marker expression is higher in the latter than in the former. This interesting finding can also be attributed to the serum-free conditions in which BV2 cells were cultured prior to the analysis. Furthermore, transcription factor *Stat1* was shown to be driving the inflammatory process and subsequent neurological dysfunction upon traumatic brain injury (TBI). However, upon tamoxifen-induced inhibition of *Stat1* in microglia and macrophages, a significant reduction in pro-inflammatory mechanisms was observed [27]. Taken together, these findings suggest that the custom microglia panel can identify expression patterns that can reflect various phenotypical changes associated with microglia.

By utilizing publicly available data from the GEO database, we have analyzed whether the genes in the custom NanoString panel can reliably display the functional changes associated with microglia when they are treated with various factors. In order to achieve this, we have used two studies. One is an RNA-seq analysis of primary microglia and BV2 cells treated with LPS for 4 h, while the other is a microarray analysis of primary microglia treated with TGFβ1 for 24 h. Several studies have shown that LPS leads to reduced expression of homeostatic markers such as *Cx3cr1*, *P2ry12*, and *Tgfbr1* while increasing markers such as *Nlrp3*, *Nos2*, and *Stat1* both in vitro and in vivo [28,29,30]. In accordance with this, our analysis of the genes in the custom panel showed that LPS treatment induces a reactive phenotype in microglia, as demonstrated by the reduction in *Hexb*, *P2ry12*, *Tgfbr1*, *Cx3cr1*, and *Olfml3* in both cell types. Interestingly, the comparison of control groups further supported the observation of increased expression of homeostatic markers in pMG cells compared to BV2 cells under the control conditions observed in the current study. These changes point to the versatility of the microglia custom panel in identifying the nuances in gene expression patterns in vitro. This is especially important as the recent review of the microglia nomenclature by experts advised disregarding the monochromatic view of M1 and M2 microglia while also recommending new terminology for the microglia response to various stimuli as well as under homeostatic conditions [31].

It has been shown that TGFβ1 is essential for postnatal microglia maturation, as it precedes the induction of microglia-specific gene expression, and loss of TGFβ signaling in adult microglia results in upregulation of microglia reactive markers, highlighting the importance of TGFβ signaling in regulating microglia functions [16,32]. Moreover, TGFβ signaling has been shown to be important in inducing microglia-specific gene signatures [7]. Consistent with this, analysis of panel genes with microarray data showed an increase in the expression of homeostatic markers and a reduction in reactive markers in TGFβ1-treated microglia. This confirms the idea that TGFβ1 treatment promotes a homeostatic microglial phenotype. Taken together, these findings suggest that our custom gene panel can adequately account for the functional changes in microglia in response to different stimuli. These results have important implications for understanding the dynamic nature of microglial function and provide a valuable resource for future functional studies in this field.

In conclusion, our study has shown that our NanoString-based microglia panel is highly effective in assessing microglial populations based on their gene expression profiles. It was also successful in validating the differences in the gene expression patterns between primary microglia and cell lines. Furthermore, this approach is not only cost-effective but also allows for the simultaneous analysis of multiple genes that can aid in understanding microglial biology. Moreover, since the nCounter system does not require high RNA input and can efficiently deal with low-quality RNA, our panel can be used on formalin-fixed paraffin-embedded samples, adding to its flexibility. Furthermore, in the future, it can be used to perform comparative studies on disease models to identify molecular signatures associated with different pathological conditions.

## Figures and Tables

**Figure 1 cells-13-00630-f001:**
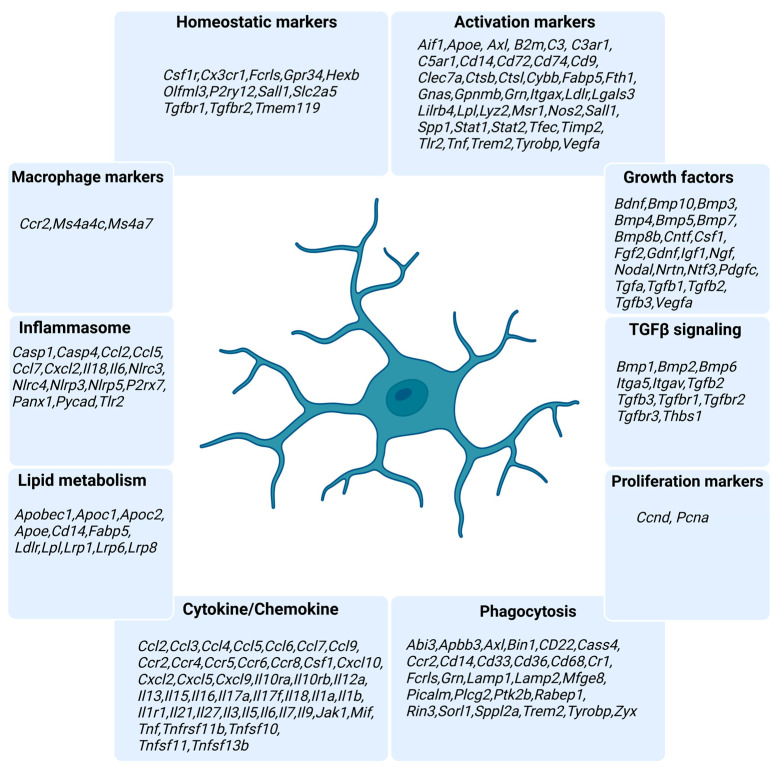
List of the genes present in the NanoString custom panel along with their annotated functions. Created with BioRender.com (accessed on 18 March 2024).

**Figure 3 cells-13-00630-f003:**
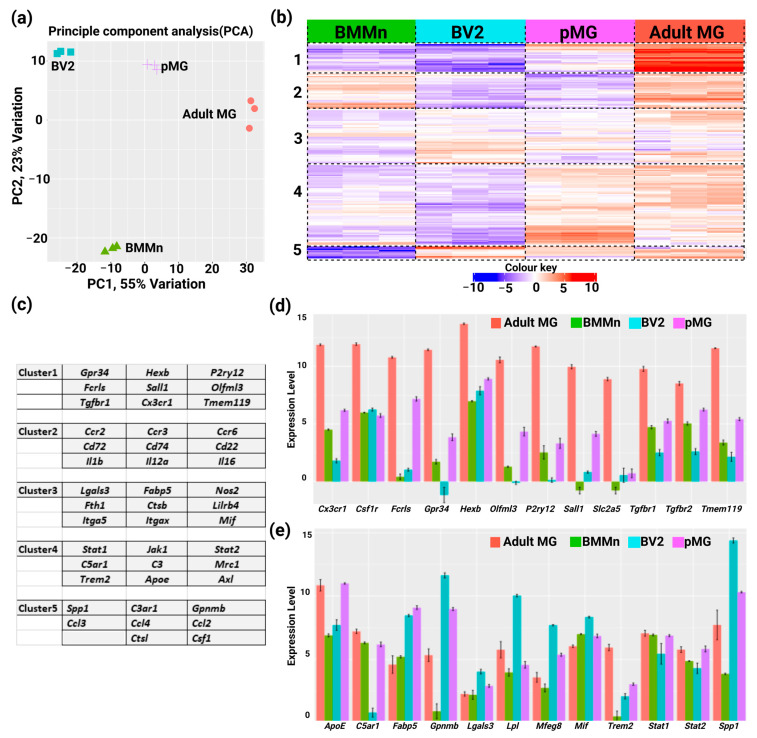
Identification of microglia gene signatures using a custom panel. (**a**) Principal Component Analysis (PCA) plot of each RNA sample colored by the experimental group; (**b**) Heatmap and K-means clustering analysis of genes present in the panel. Normalized counts were centered and clustered by Pearson correlation. The color key corresponds to the Z-score employed to identify distinct gene expression patterns in Adult MG, BMMn cells, pMG, and BV2 cells; (**c**) Table with microglia-relevant genes present in each cluster of the heatmap; (**d**,**e**) Plots representing the expression patterns of homeostatic markers and reactive markers, respectively. Error bars represent the SEM.

**Figure 4 cells-13-00630-f004:**
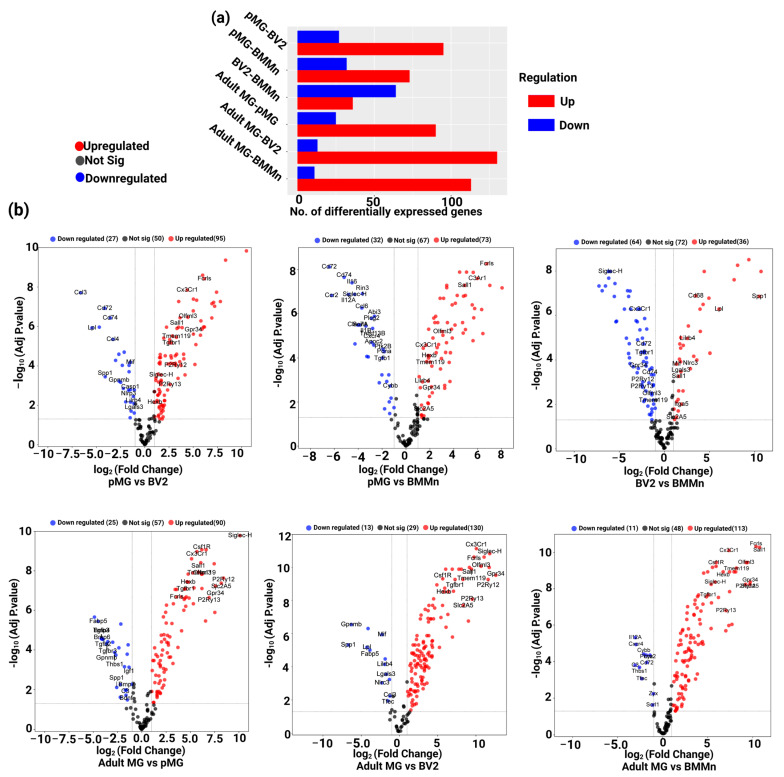
Analysis of differential gene expression: (**a**) Overview of differential gene expression analysis (DEG) using iDEP for all possible cell type comparisons in this study; (**b**) Volcano plots of DEG analysis. Upper panel: DEG analysis comparing pMG to BV2 cells, BMMns, and BV2 microglia compared to BMMns. Bottom panel: DEG analysis comparing adult MG to pMG, BV2 cells, and BMMns.

**Figure 5 cells-13-00630-f005:**
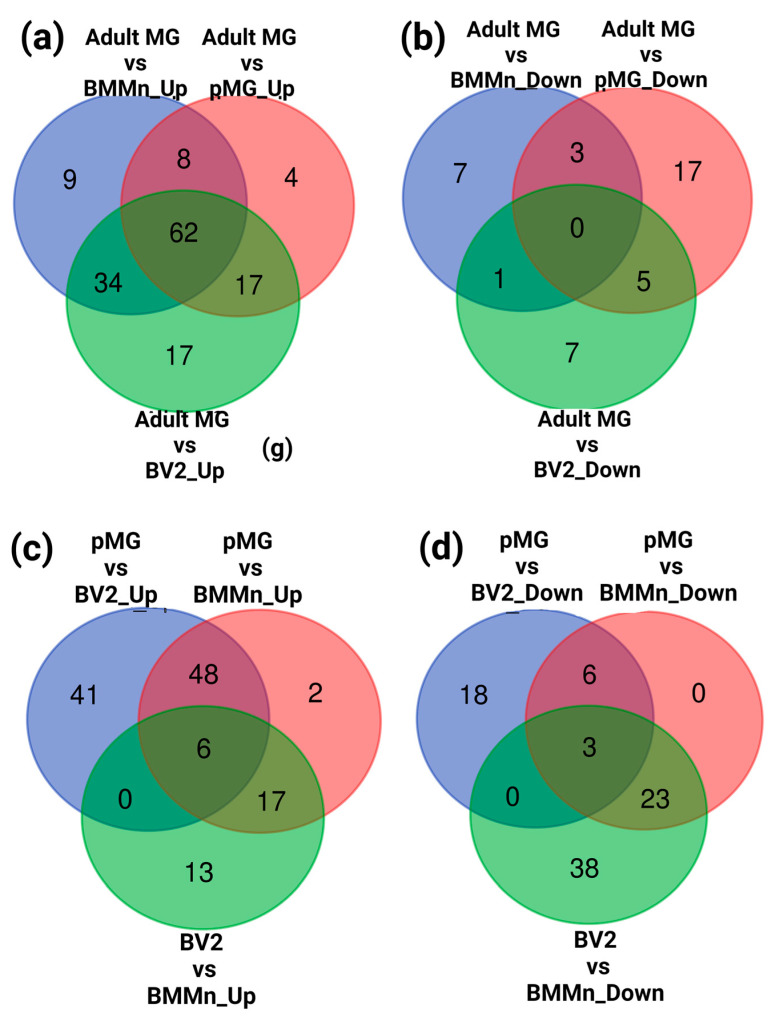
Analysis of common and cell-specific changes in gene expression. (**a**) Venn diagram illustrating the number of upregulated genes in adult MG compared to pMG, BV2, and BMMn cells along with the overlap in the expression pattern; (**b**) Venn diagram of the downregulated genes for the similar comparison as in (**a**); (**c**) Venn diagrams depicting the number of upregulated genes in pMG when compared to BV2 cells and BMMns, and in BV2 cells vs. BMMns along with overlapped genes; (**d**) Venn diagram of the downregulated genes for the similar comparison as in (**c**).

**Figure 6 cells-13-00630-f006:**
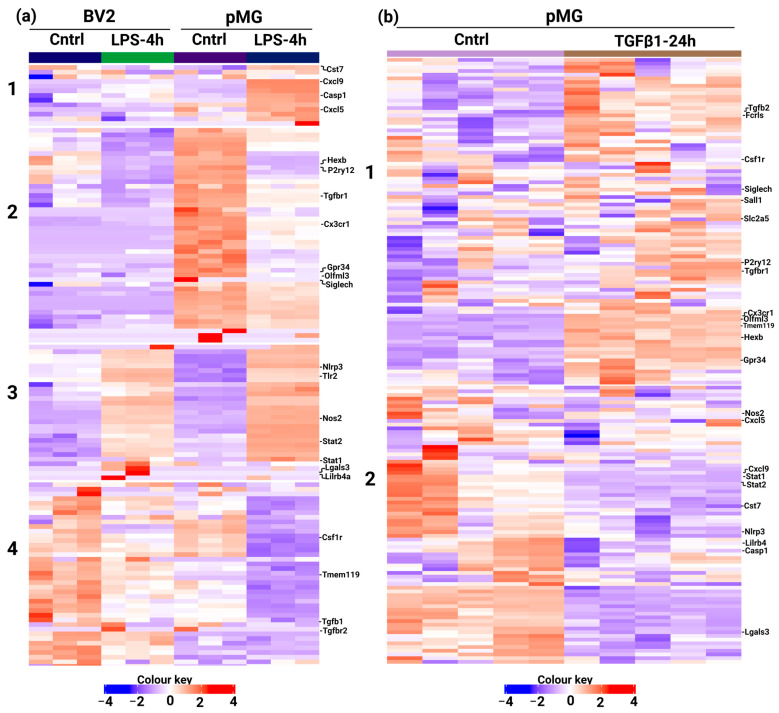
Microglia custom panel can reflect the state-specific changes associated with different treatments: (**a**) Heatmap and K-means clustering analysis of genes present in the panel from GSE79898 (Cntrl vs. LPS in BV2 cells) and GSE803304 (Cntrl vs. LPS in pMG cells). Normalized counts were centered and clustered by Pearson correlation. The color key corresponds to row Z-score employed; (**b**) Heatmap and K-means clustering analysis of genes present in the panel from GSE115652 of the pMG treated with TGFβ1 for 24 h. Normalized counts were centered and clustered by Pearson correlation. The color key corresponds to the row Z-score employed.

## Data Availability

The data that support the findings of this study are available from the corresponding author upon reasonable request.

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
