# Peer review of "A Custom Panel for Profiling Microglia Gene Expression"

_cells, 2024, doi:10.3390/cells13070630_

Round 1

Reviewer 1 Report

Comments and Suggestions for Authors

The manuscript entitle “A Custom Panel for Profiling Microglia Gene Expression” investigates the functionality multiplexed approach to quantify the microglia gene expression. For that, they analyzed the RNA expression from bone marrow monocytes, BV2 microglia cell line, primary mouse microglia, and microglia isolated from adult mice using their custom panel. The ultimate objective of the work was to demonstrate that employing a panel embedded with specific markers of microglia would not only reduce overall costs but also facilitate its future utilization in comparisons between normal models and those exhibiting different pathologies. Although the findings described in the manuscript are interesting the article requires extensive review before it can be published.:

Major points:

1.       One of the main major points that the authors should update Microglia nomenclature. The authors should know this issue with microglial nomenclature and cite the recent consensus paper of 94 authors published in Neuron 2022 recommending that different microglia terms are banished from the literature. The paper is plagued with this misconception and the authors should update their terminology.

2.       The subsection 3.1 should be rewrite since the review has not been able understand it. In the text the author talks about two principal components (PC1 and PC2), however in the figure only PC is shown. Late, the talk about six cluster but in the figure 3 only mention five.

3.       In figure 3d and e, it would be possible to perform statistical analysis? The graph indicates expression level, is it in comparison with a housekeeping gene? If so, they must indicate what and how they did it.

Minor points:

There are many formal errors in the writing, extra and missing spaces for example. They should review it thoroughly.

Author Response

The manuscript titled “A Custom Panel for Profiling Microglia Gene Expression” investigates the functionality multiplexed approach to quantify the microglia gene expression. For that, they analyzed the RNA expression from bone marrow monocytes, BV2 microglia cell line, primary mouse microglia, and microglia isolated from adult mice using their custom panel. The ultimate objective of the work was to demonstrate that employing a panel embedded with specific markers of microglia would not only reduce overall costs but also facilitate its future utilization in comparisons between normal models and those exhibiting different pathologies. Although the findings described in the manuscript are interesting the article requires extensive review before it can be published.:

Major points:

  1. One of the main major points that the authors should update Microglia nomenclature. The authors should know this issue with microglial nomenclature and cite the recent consensus paper of 94 authors published in Neuron 2022 recommending that different microglia terms are banished from the literature. The paper is plagued with this misconception and the authors should update their terminology.

    We thank the reviewer for this important comment and acknowledging this very important paper (Paolicelli et al. 2022). We have used the updated terminology in most of the cases in the manuscript. However, the term ``activation markers`` was used 3 times in the manuscript to refer to some genes that are increased under inflammatory conditions, for instance. It has now been changed to ”reactive”. The new revised nomenclature also suggests the usage of ``Homeostatic microglia`` instead of resting microglia. This term has been used to refer to the microglia under ``normal`` conditions throughout. Although, as suggested by Paolicelli et al, it is important to understand different subsets within the homeostatic microglia depending on which homeostatic markers are expressed, this question is beyond the scope of the current manuscript. However, we were unable to cite the paper in the original manuscript as there is no reference to updated nomenclature. 

  2. The subsection 3.1 should be rewrite since the review has not been able understand it. In the text the author talks about two principal components (PC1 and PC2), however in the figure only PC is shown. Late, the talk about six cluster but in the figure 3 only mention five. 

We thank the reviewer for bringing this to our notice. Indeed, the PCA analysis should read PC1 vs PC2. This typing mistake is now rectified. Regarding the clusters, it was mistakenly written as 6 clusters. There were only 5 meaningful clusters. An elbow plot used for determining the k-means clusters is now added as a supplementary figure (Figure S1). Information regarding this has been updated in the results part in line 179.

  1. In figure 3d and e, it would be possible to perform statistical analysis? The graph indicates expression level, is it in comparison with a housekeeping gene? If so, they must indicate what and how they did it.

We thank the reviewer for the question. Figures 3d and 3e are directly derived from iDEP using the gene plot option in the pre-processing tab. The normalization was performed against a set of 5 housekeeping genes present in the panel. Statistical analysis using graph pad prism resulted in an image with a lot of significant changes filled with asterisks, which is difficult to read. Moreover, the plot form iDEP was directly used in the manuscript so as not to modify the statistics performed by iDEP.  However, we have added a supplementary file with the normalized counts from which individual statistics can be performed. Moreover, we have now added an additional supplementary file with the graphs and table of 2-way ANOVA analysis for the genes in Figures 3d and e.

Minor points:

There are many formal errors in the writing, extra and missing spaces for example. They should review it thoroughly.
The manuscript has been thoroughly revised for any formal errors.

Paolicelli, Rosa C., Amanda Sierra, Beth Stevens, Marie-Eve Tremblay, Adriano Aguzzi, Bahareh Ajami, Ido Amit, et al. 2022. ‘Microglia States and Nomenclature: A Field at Its Crossroads’. Neuron 110 (21): 3458–83. https://doi.org/10.1016/j.neuron.2022.10.020.

Reviewer 2 Report

Comments and Suggestions for Authors

The manuscript by Potru and colleagues compares markers assigned to microglia at different phases and, using Nanostring technology, determines a specific group of genes that can be used to profile microglia. The concept is interesting, but there are concerns about the presentation of the manuscript that significantly decrease enthusiasm. Specifically:

- They are referring to figure 3a - not 1a - in line 181. 

- They also discuss 6 clusters, but it is hard to see 6 clusters anywhere. There are 4 clusters on the PCA in fig 3a (which is mislabeled, you cannot plot pc1 vs pc1. It has to be plotted vs pc2). But then in 3b and c, they have 5 clusters, and in the text they don’t even describe a cluster 6, they stop at 5.

- Why are the authors using NMRI (outbred) mice for perinatal and adult microglia, but are using 57Bl6 adult brains?

- What is a biological significance of using the BV2 cells?

Author Response

The manuscript by Potru and colleagues compares markers assigned to microglia at different phases and, using Nanostring technology, determines a specific group of genes that can be used to profile microglia. The concept is interesting, but there are concerns about the presentation of the manuscript that significantly decrease enthusiasm. Specifically:

  1. They are referring to figure 3a - not 1a - in line 181. 

We apologize for this mistake and thank the reviewer for bringing this to our attention. Correct figure number is now added and was changed to Figure 3a. 

  1. They also discuss 6 clusters, but it is hard to see 6 clusters anywhere. There are 4 clusters on the PCA in fig 3a (which is mislabeled, you cannot plot pc1 vs pc1. It has to be plotted vs pc2). But then in 3b and c, they have 5 clusters, and in the text they don’t even describe a cluster 6, they stop at 5.

We apologize for this typing error. PCA analysis was indeed performed for PC1 vs PC2. This typing mistake is now rectified. Regarding the clusters, it was mistakenly written as 6 clusters. There were only 5 meaningful clusters. An elbow plot used for determining the k-means clusters is now added as a supplementary figure (Figure S1). Information regarding this has been updated in the results part in line 179.

  1. Why are the authors using NMRI (outbred) mice for perinatal and adult microglia, but are using 57Bl6 adult brains?

We thank the reviewer for this interesting question. We have been routinely using NMRI mice for the regular primary cultures. Even though there is no added biological relevance for microglia cells themselves, this is purely a technical concern as NMRI mice, in our experience, tend to have bigger litters that help us culturing higher number of microglia cells. However, it should be noted that the differences between microglia from black6 and NMRI mice were previously studied and no changes were reported between them (Zlotnik and Spittau, 2014).

  1. What is a biological significance of using the BV2 cells?

BV2 cells are one of the most widely used murine microglia cell line. Even though they share some similarities with primary microglia and ex vivo microglia, the results from these cells should be considered carefully as they were shown to be different from microglia even at transcriptomic level (Das et al., 2016). The rationale behind including BV2 cells in the current study is to emphasize this difference and to show that the genes in custom panel can efficiently these differences. However, BV2 cells are still relevant model to perform pilot studies and to get an understanding of microglia behaviour. However, these findings should subsequently be validated using further in vitro (primary microglia) and in vivo microglia cells. More context on the changes BV2 cells display when compared to other cell types in also given in the results part for Figure 5.

Das, A., Kim, S.H., Arifuzzaman, S., Yoon, T., Chai, J.C., Lee, Y.S., Park, K.S., Jung, K.H., Chai, Y.G., 2016. Transcriptome sequencing reveals that LPS-triggered transcriptional responses in established microglia BV2 cell lines are poorly representative of primary microglia. J. Neuroinflammation 13, 182. https://doi.org/10.1186/s12974-016-0644-1

Zlotnik, A., Spittau, B., 2014. GDNF fails to inhibit LPS-mediated activation of mouse microglia. J. Neuroimmunol. 270, 22–28. https://doi.org/10.1016/j.jneuroim.2014.03.006

Reviewer 3 Report

Comments and Suggestions for Authors

Summary

The study submitted by Potru and Colleagues describes a custom-made gene expression panel that was subsequently applied to profile microglia from primary cultures, adult mice as well as the commonly used BV2 microglia cell line and bone marrow – derived monocytes. They detected transcriptional differences between these cells including microglia signature genes, which were found to be expressed at higher levels in adult microglia and were barely detectable in BV2 cells. Finally, they demonstrate that their gene panel is capable of detecting changes in murine microglia that can occur under pathological conditions by analyzing publicly available gene expression data from LPS-challenged and TGFß-treated microglia.

Overall, given the broad interest of the microglia field in gene expression studies and the high costs of RNA-seq approaches, this study is of broad interest and describes an interesting method to characterize microglia gene expression. The experiments are conclusively described and the conclusions drawn by the authors are, in general, supported by the data they present. Nevertheless, there are some points of concern that mainly deal with the manuscript and should be addressed before acceptance for publication.

Major Points

1)        The authors describe which genes are included in their gene expression panel and mention the biological processes covered by these genes. However, the authors should also explain which criteria were applied to select these target genes including the studies which served as the foundation for including these markers into their gene panel. This explanation would strongly improve the understanding of the rationale that lies behind the genes chosen for this custom panel and should be either included in the Materials and Methods section or at the beginning of the Results section (in that case, it would make sense to reference Figure 1 which illustrates the different genes that are covered by this panel).

2)        The authors should consider to split Figure 4 into two separate figures as it currently occupies two pages. One way to effectively separate the data into two figures is to put Figure 4a and b into one figure and the Venn plots and corresponding gene lists into a second one. Alternatively, the Venn plots can be put below the volcano plots in figure 4 whereas the gene lists may be put into a separate supplemental figure for the interested reader.  

Minor Points

1)        In line 26-27, the authors state that microglia are the sentinel immune cells that reside within the CNS, which is, of course, absolutely correct. However, I suggest to include the fact that microglia are parenchymal, tissue-resident macrophages. This would emphasize the difference between microglia and extraparenchymal CAMs in the perivascular spaces, meninges etc. that also belong to the innate immune system of the CNS. 

2)        The authors should check the gene names utilized in line 37-40. For example, Sall1 is named “spalt like transcription factor 1”, according to the MGI database. 

3)        In line 76 and 118, the authors state that adult microglia were isolated from C57BL/6 mice – can the authors provide information from which subline these cells were isolated (e.g. C57BL/6J or C57BL/6N mice)?

4)        Refering to the description of the flow cytometry procedure in the Materials and Methods section, the authors did not mention a Fc-receptor blocking step. Did the MACS protocol already contain a Fc-receptor blocking step? 

5)        The authors should check the referencing of the figures throughout the manuscript. Exemplarily, figure 2a is not referenced in the text (at least, I couldn’t find it). Furthermore, in chapter 3.1, the authors reference Figure 1a-e, which should be corrected as Figure 3a-e are mentioned, from my point of view. 

6)        Regarding the description of the results of the differential gene expression analysis, beginning with line 211, the authors should not only include the numbers of up- and downregulated genes, but also discuss which genes are up- and downregulated, as shown in the Volcano plots. Exemplarily, the volcano plot comparing adult microglia with BV2 cells clearly demonstrates the upregulation of microglia signature genes like Tmem119, P2ry12 etc.. This is noteworthy as it strongly supports, from my point of view, one of the main messages of the study.  

7)        In line 244, the authors state that 48 genes are commonly upregulated when comparing pMG and BV2 cells. This is misleading as, according to the Venn diagram, 48 genes are equally upregulated in the comparisons pMG vs BV2 and pMG vs. Bone Marrow Monocytes. The authors should check this. 

8)        In general, the authors should check the manuscript for the writing of numbers in the manuscript. In some cases, the numbers are spelled out (e.g. line 245, 265), whereas in the most cases, they are not. This should be consistent across the whole manuscript.

9)        In line 275, Hexb and P2ry12 should be written in italic.

10)      In line 276, the authors mentioned Gpr35. I assume that Gpr34 is meant?

11)      In line 283, “… etc. were upregulated,” is doubled – the authors should check this. 

12)      In line 305, the sentence should be rephrased as in vivo microglia display an homeostatic phenotype.

13)      In line 328, the authors state that inhibition of TGFß-signalling reduces Mrc1-expression, which is known to be a CAM-marker in the CNS. However, when looking at the title of the referenced publication, it seems that the opposite is the case as Mrc1 is increased upon silencing TGFß-signaling. The authors should clarify this. 

14)      In line 371 and 375, the full stop in front of “Furthermore” is missing. 

Comments on the Quality of English Language

Generally, the overall quality of the language is good and the points made by the authors can be understood, however, some minor points such as the spelling of numbers, as mentioned in the comments section already, need to be addressed before acceptance. 

Author Response

Reviewer 3

The study submitted by Potru and Colleagues describes a custom-made gene expression panel that was subsequently applied to profile microglia from primary cultures, adult mice as well as the commonly used BV2 microglia cell line and bone marrow – derived monocytes. They detected transcriptional differences between these cells including microglia signature genes, which were found to be expressed at higher levels in adult microglia and were barely detectable in BV2 cells. Finally, they demonstrate that their gene panel is capable of detecting changes in murine microglia that can occur under pathological conditions by analyzing publicly available gene expression data from LPS-challenged and TGFß-treated microglia.

Overall, given the broad interest of the microglia field in gene expression studies and the high costs of RNA-seq approaches, this study is of broad interest and describes an interesting method to characterize microglia gene expression. The experiments are conclusively described and the conclusions drawn by the authors are, in general, supported by the data they present. Nevertheless, there are some points of concern that mainly deal with the manuscript and should be addressed before acceptance for publication.

Major Points

1)        The authors describe which genes are included in their gene expression panel and mention the biological processes covered by these genes. However, the authors should also explain which criteria were applied to select these target genes including the studies which served as the foundation for including these markers into their gene panel. This explanation would strongly improve the understanding of the rationale that lies behind the genes chosen for this custom panel and should be either included in the Materials and Methods section or at the beginning of the Results section (in that case, it would make sense to reference Figure 1 which illustrates the different genes that are covered by this panel).

The genes for the custom panel have been chosen according to recent studies defining microglia markers in health and disease and markers that distinguish microglia from monocytes/macrophages (Keren-Shaul et al. 2017; Butovsky and Weiner 2018; Grubman et al. 2021; Haage et al. 2019; Zrzavy et al. 2017; Krasemann et al. 2017; Ayata et al. 2018).These references are now added in the results part before going into the explanation of figure 3.

2)        The authors should consider to split Figure 4 into two separate figures as it currently occupies two pages. One way to effectively separate the data into two figures is to put Figure 4a and b into one figure and the Venn plots and corresponding gene lists into a second one. Alternatively, the Venn plots can be put below the volcano plots in figure 4 whereas the gene lists may be put into a separate supplemental figure for the interested reader.  

Figure 4 is now split into two parts. The Venn diagrams are now moved to Figure 5 while the tables with the gene lists were placed in a supplementary file and referenced in the text.

Minor Points

  • In line 26-27, the authors state that microglia are the sentinel immune cells that reside within the CNS, which is, of course, absolutely correct. However, I suggest to include the fact that microglia are parenchymal, tissue-resident macrophages. This would emphasize the difference between microglia and extraparenchymal CAMs in the perivascular spaces, meninges etc. that also belong to the innate immune system of the CNS. 

We thank the reviewer and agree with the fact that microglia are indeed parenchymal macrophages in CNS and should be distinguished from other immune populations such as dendritic cells, macrophages and mast cells that are usually present in the linings, such as the meninges and choroid plexus.

  • The authors should check the gene names utilized in line 37-40. For example, Sall1 is named “spalt like transcription factor 1”, according to the MGI database. 

We thank the reviewer for pointing this out. Sall1 in mice is indeed ``spalt like transcription factor 1``. We have now changed the description in text.

3)        In line 76 and 118, the authors state that adult microglia were isolated from C57BL/6 mice – can the authors provide information from which subline these cells were isolated (e.g. C57BL/6J or C57BL/6N mice)?

We apologize for not mentioning the exact subline that we have used. For clarification, the mice subline from which we isolated the microglia is C57BL/6J and this information has been added to the text.

4)        Referring to the description of the flow cytometry procedure in the Materials and Methods section, the authors did not mention a Fc-receptor blocking step. Did the MACS protocol already contain a Fc-receptor blocking step? 

We agree with the reviewer that it is usually important to perform a Fc-receptor blocking step before staining with different antibodies to reduce background signaling. However, an additional Fc-receptor blocking step is not necessary in this case because we used REAfinity antibodies from Miltenyi biotec, which are specifically designed for flow cytometry. To eliminate the background, the Fc-region was mutated. According to the supplier, no Fc-receptor blocking step is required.

5)        The authors should check the referencing of the figures throughout the manuscript. Exemplarily, figure 2a is not referenced in the text (at least, I couldn’t find it). Furthermore, in chapter 3.1, the authors reference Figure 1a-e, which should be corrected as Figure 3a-e are mentioned, from my point of view. 

We thank the reviewer for the question. It is true that figure 2a is not referenced as it is used a quick summary image for the all the methods used in the paper. Therefore, it was not mentioned in the text as it would have to be referenced time and again in the methods part.

We apologize for the mistake in referencing the images. We have now included a correct referencing to the results in the Figure 3 and, the whole manuscript was cross checked for similar errors.

6)        Regarding the description of the results of the differential gene expression analysis, beginning with line 211, the authors should not only include the numbers of up- and downregulated genes, but also discuss which genes are up- and downregulated, as shown in the Volcano plots. Exemplarily, the volcano plot comparing adult microglia with BV2 cells clearly demonstrates the upregulation of microglia signature genes like Tmem119, P2ry12 etc.. This is noteworthy as it strongly supports, from my point of view, one of the main messages of the study.  

We thank the reviewer for this question and agree with the importance of mentioning the genes.

We have now added the information regarding the up and downregulated genes from the volcano plots. Also, an emphasis was put on the changes in BV2 cells when compared to the other cells used in the study.

7)        In line 244, the authors state that 48 genes are commonly upregulated when comparing pMG and BV2 cells. This is misleading as, according to the Venn diagram, 48 genes are equally upregulated in the comparisons pMG vs BV2 and pMG vs. Bone Marrow Monocytes. The authors should check this. 

We thank the reviewer for this observation. The text is now corrected to reflect the Venn diagrams.

8)        In general, the authors should check the manuscript for the writing of numbers in the manuscript. In some cases, the numbers are spelled out (e.g. line 245, 265), whereas in the most cases, they are not. This should be consistent across the whole manuscript.

We thank the reviewer for bringing this to our attention. Numbers below 10 are now spelled out across the manuscript while the numbers above 10 are given as numbers.

9)        In line 275, Hexb and P2ry12 should be written in italic.

Fonts of Hexb and P2ry12 is changed to italic

10)      In line 276, the authors mentioned Gpr35. I assume that Gpr34 is meant?

Gpr35 is now changed to Gpr34. 

11)      In line 283, “… etc. were upregulated,” is doubled – the authors should check this.

We apologize for this overlook and the text is rectified.

12)      In line 305, the sentence should be rephrased as in vivo microglia display an homeostatic phenotype.

Ex vivo in line 305 is now changed to in vivo as per the suggestion of the reviewer.

13)      In line 328, the authors state that inhibition of TGFß-signalling reduces Mrc1-expression, which is known to be a CAM-marker in the CNS. However, when looking at the title of the referenced publication, it seems that the opposite is the case as Mrc1 is increased upon silencing TGFß-signaling. The authors should clarify this. 

We thank the reviewer for this very important comment. Mrc1 is increased upon the inhibition or downregulation of Tgfβ signaling. In this instance, reduced is written accidentally instead of increased. We have now changed the text to reflect the correct change in Mrc1 expression as per the previously published study.

14)      In line 371 and 375, the full stop in front of “Furthermore” is missing. 

      Full stops are now added at both places.

Comments on the Quality of English Language

Generally, the overall quality of the language is good and the points made by the authors can be understood, however, some minor points such as the spelling of numbers, as mentioned in the comments section already, need to be addressed before acceptance. 

  1. Ayata, Pinar, Ana Badimon, Hayley J. Strasburger, Mary Kaye Duff, Sarah E. Montgomery, Yong-Hwee E. Loh, Anja Ebert, et al. 2018. ‘Epigenetic Regulation of Brain Region-Specific Microglia Clearance Activity’. Nature Neuroscience 21 (8): 1049–60. https://doi.org/10.1038/s41593-018-0192-3.
  2. Butovsky, Oleg, and Howard L. Weiner. 2018. ‘Microglial Signatures and Their Role in Health and Disease’. Nature Reviews. Neuroscience 19 (10): 622–35. https://doi.org/10.1038/s41583-018-0057-5.
  3. Grubman, Alexandra, Xin Yi Choo, Gabriel Chew, John F. Ouyang, Guizhi Sun, Nathan P. Croft, Fernando J. Rossello, et al. 2021. ‘Transcriptional Signature in Microglia Associated with Aβ Plaque Phagocytosis’. Nature Communications 12 (1): 3015. https://doi.org/10.1038/s41467-021-23111-1.
  4. Haage, Verena, Marcus Semtner, Ramon Oliveira Vidal, Daniel Perez Hernandez, Winnie W. Pong, Zhihong Chen, Dolores Hambardzumyan, et al. 2019. ‘Comprehensive Gene Expression Meta-Analysis Identifies Signature Genes That Distinguish Microglia from Peripheral Monocytes/Macrophages in Health and Glioma’. Acta Neuropathologica Communications 7 (1): 20. https://doi.org/10.1186/s40478-019-0665-y.
  5. Keren-Shaul, Hadas, Amit Spinrad, Assaf Weiner, Orit Matcovitch-Natan, Raz Dvir-Szternfeld, Tyler K. Ulland, Eyal David, et al. 2017. ‘A Unique Microglia Type Associated with Restricting Development of Alzheimer’s Disease’. Cell 169 (7): 1276-1290.e17. https://doi.org/10.1016/j.cell.2017.05.018.
  6. Krasemann, Susanne, Charlotte Madore, Ron Cialic, Caroline Baufeld, Narghes Calcagno, Rachid El Fatimy, Lien Beckers, et al. 2017. ‘The TREM2-APOE Pathway Drives the Transcriptional Phenotype of Dysfunctional Microglia in Neurodegenerative Diseases’. Immunity 47 (3): 566-581.e9. https://doi.org/10.1016/j.immuni.2017.08.008.
  7. Zrzavy, Tobias, Simon Hametner, Isabella Wimmer, Oleg Butovsky, Howard L. Weiner, and Hans Lassmann. 2017. ‘Loss of “homeostatic” Microglia and Patterns of Their Activation in Active Multiple Sclerosis’. Brain: A Journal of Neurology 140 (7): 1900–1913. https://doi.org/10.1093/brain/awx113.

Round 2

Reviewer 1 Report

Comments and Suggestions for Authors

The authors have made most of the proposed changes. Even so, I have observed some formal failures that should be reviewed.

Furthermore, the supplementary information is not referenced in the text correctly. Also note that many things that have been changed in the text are not indicated in red, so it has been difficult to follow the changes with respect to the previous version.